# Deep Scale-spaces: Equivariance Over Scale

**Daniel E. Worrall**[*]
AMLAB, *Philips Lab*
University of Amsterdam
d.e.worrall@uva.nl

**Max Welling**
AMLAB, *Philips Lab*
University of Amsterdam
m.welling@uva.nl

## Abstract

We introduce deep scale-spaces (DSS), a generalization of convolutional networks, exploiting the scale symmetry structure of conventional image recognition tasks. Put plainly, the class of an image is invariant to the scale at which it is viewed. We construct scale equivariant cross-correlations based on a principled extension of convolutions, grounded in the theory of scale-spaces and semigroups. As a very basic operation, these cross-correlations can be used in almost any modern deep learning architecture in a plug-and-play manner. We demonstrate our networks on the Patch Camelyon and Cityscapes datasets, to prove their utility and perform introspective studies to further understand their properties.

## 1 Introduction

Scale is inherent in the structure of the physical world around us and the measurements we make of it. Ideally, the machine learning models we run on this perceptual data should have a notion of scale, which is either learnt or built directly into them. However, the state-of-the-art models of our time, convolutional neural networks (CNNs) [Lecun et al., 1998], are predominantly local in nature due to small filter sizes. It is not thoroughly understood how they account for and reason about multiscale interactions in their deeper layers, and empirical evidence [Chen et al., 2018, Yu and Koltun, 2015, Yu et al., 2017] using dilated convolutions suggests that there is still work to be done in this arena.

In computer vision, typical methods to circumvent scale are: *scale averaging*, where multiple scaled versions of an image are fed through a network and then averaged [Kokkinos, 2015]; *scale selection*, where an object's scale is found and local computations are adapted accordingly [Girshick et al., 2014, Shelhamer et al., 2019]; and *scale augmentation*, where multiple scaled versions of an image are added to the training set [Barnard and Casasent, 1991]. While these methods help, they lack explicit mechanisms to fuse information from different scales into the same representation. Many works do indeed follow this approach Ke et al. [2017], Saxena and Verbeek [2016], Lin et al. [2017], Kanazawa et al. [2014], Huang et al. [2018] and in this work, we follow this line of thinking and construct a generalized convolution taking, as input, information from different scales.

The utility of convolutions arises in scenarios where there is a *translational symmetry* (translation invariance) inherent in the task of interest [Cohen and Welling, 2016a]. Examples of such tasks are object classification [Krizhevsky et al., 2012], object detection [Girshick et al., 2014], or dense image labelling [Long et al., 2015]. By using translational weight-sharing [Lecun et al., 1998] for these tasks, we reduce the parameter count while preserving symmetry in the deeper layers. The overall effect is to improve sample complexity and thus reduce generalization error [Sokolic et al., 2017]. Furthermore, it has been shown that convolutions (and various reparameterizations of them) are the only linear operators that preserve symmetry [Kondor and Trivedi, 2018]. Attempts have been made to extend convolutions to scale, but they either suffer from breaking translation symmetry [Henriques and Vedaldi, 2017, Esteves et al., 2017], making the assumption that scalings can be modelled in the same way as rotations [Marcos et al., 2018], or ignoring symmetry constraints [Hilbert et al., 2018].

---

[*]deworrall92.github.io

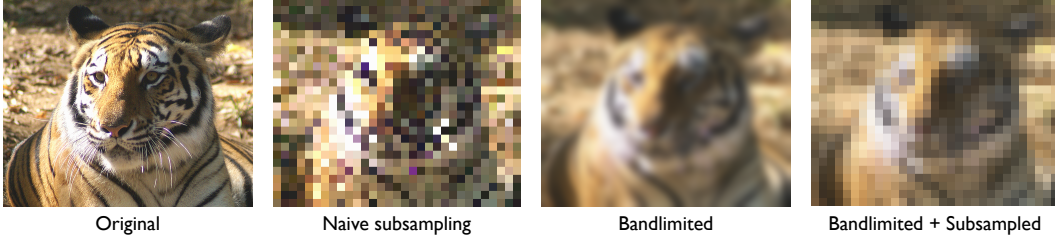

| Original | Naive subsampling | Bandlimited | Bandlimited + Subsampled |

Figure 1: How to correctly downsample an image. Left to right: The original high-resolution image; an ×1/8 subsampled image, notice how a lot of the image structure has been destroyed; a high-resolution, bandlimited (blurred) image; a bandlimited and ×1/8 subsampled image. Compare the bandlimited and subsampled image with the naïvely subsampled image. Much of the low-frequency image structure is preserved in the bandlimited and subsampled image. Image source: ImageNet.

The problem with the aforementioned approaches is that they fail to account for the unidirectional nature of scalings. In data there exist many one-way transformations, which cannot be inverted. Examples are occlusions, causal translations, downscalings of discretized images, and pixel lighting normalization. In each example the transformation deletes information from the original signal, which cannot be regained, and thus it is non-invertible. We extend convolutions to these classes of symmetry under noninvertible transformations via the theory of semigroups. Our contributions are the introduction of a semigroup equivariant correlation and a scale-equivariant CNN.

## 2 Background

This section introduces some key concepts such as groups, semigroups, actions, equivariance, group convolution, and scale-spaces. These concepts are presented for the benefit of the reader, who is not expected to have a deep knowledge of any of these topics *a priori*.

**Downsizing images**  We consider sampled images $f \in L^2(\mathbb{Z}^2)$ such as in Figure 1. For image $f$, $x$ is pixel position and $f(x)$ is pixel intensity. If we wish to downsize by a factor of 8, a naïve approach would be to subsample every $8^{\text{th}}$ pixel: $f_{\text{down}}(x) = f(8x)$. This leads to an artifact, *aliasing* [Mallat, 2009, p.43], where the subsampled image contains information at a higher-frequency than can be represented by its resolution. The fix is to *bandlimit* pre-subsampling, suppressing high-frequencies with a blur. Thus a better model for downsampling is $f_{\text{down}}(x) = [G *_{\mathbb{Z}^d} f](8x)$, where $*_{\mathbb{Z}^d}$ denotes convolution over $\mathbb{Z}^d$, and $G$ is an appropriate blur kernel (discussed later). Downsizing involves necessary information loss and cannot be inverted [Lindeberg, 1997]. Thus upsampling of images is not well-defined, since it involves imputing high-frequency information, not present in the low resolution image. As such, *in this paper we only consider image downscaling*.

**Scale-spaces**  Scale-spaces have a long history, dating back to the late fifties with the work of Iijima [1959]. They consist of an image $f_0 \in L^2(\mathbb{R}^d)$ and multiple blurred versions of it. Although sampled images live on $\mathbb{Z}^d$, scale-space analysis tends to be over $\mathbb{R}^d$, but many of the results we present are valid on both domains. Among all variants, the Gaussian scale-space (GSS) is the commonest [Witkin, 1983]. Given an initial image $f_0$, we construct a GSS by convolving $f_0$ with an isotropic (rotation invariant) Gauss-Weierstrass kernel $G(x, t) = (4\pi t)^{-d/2} \exp\left\{-\|x\|^2/4t\right\}$ of variable width $\sqrt{t}$ and spatial positions $x$. The GSS is the complete set of responses $f(t, x)$:

$$f(t, x) = [G(\cdot, t) *_{\mathbb{R}^d} f_0](x), \qquad t > 0 \tag{1}$$
$$f(0, x) = f_0(x), \tag{2}$$

where $*_{\mathbb{R}^d}$ denotes convolution over $\mathbb{R}^d$. The higher the level $t$ (larger blur) in the image, the more high frequency details are removed. An example of a scale-space $f(t, x)$ can be seen in Figure 2. An interesting property of scale-spaces is the *semigroup property* [Florack et al., 1992], sometimes referred to as the *recursivity principle* [Pauwels et al., 1995], which is

$$f(s + t, \cdot) = G(\cdot, s) *_{\mathbb{R}^d} f(t, \cdot) \tag{3}$$

for $s, t > 0$. It says that we can generate a scale-space from other levels of the scale-space, not just from $f_0$. Furthermore, since $s, t > 0$ it also says that we cannot generate sharper images from blurry ones, using just a Gaussian convolution. Thus moving to blurrier levels encodes a degree of information loss. This property emanates from the closure of Gaussians under convolution, namely for multidimensional Gaussians with covariance matrices $\Sigma$ and $T$

$$G(\cdot, \Sigma + T) = G(\cdot, \Sigma) *_{\mathbb{R}^d} G(\cdot, T). \qquad (4)$$

We assume the initial image $f_0$ has a maximum spatial frequency content—dictated by pixel pitch in discretized images—which we model by assuming the image has already been convolved with a width $s_0$ Gaussian, which we call the *zero-scale*. Thus an image of bandlimit $s$ in the scale-space is found at GSS slice $f(s - s_0, \cdot)$, which we see from Equation 3. There are many varieties of scale-spaces: the $\alpha$-scale-spaces [Pauwels et al., 1995], the discrete Gaussian scale-spaces [Lindeberg, 1990], the binomial scale-spaces [Burt, 1981], etc. These all have specific kernels, analogous to the Gaussian, which are closed under convolution (details in supplement).

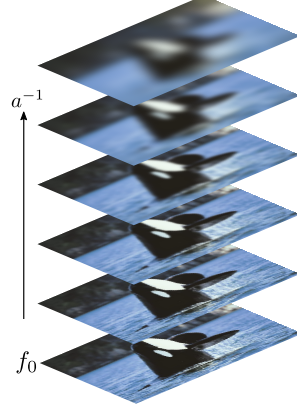

Slices at level $t$ in the GSS correspond to images downsized by a *dilation factor* $0 \leq a \leq 1$ ($1$ = no downsizing, $0$ = shrinkage to a point). An $a$-dilated and appropriately bandlimited image $p(a, x)$ is found as (details in supplement)

Figure 2: A Scale-space: For implementations we logarithmically discretize the scale-axis in $a^{-1}$.

$$p(a, x) = f(t(a, s_0), a^{-1}x), \qquad t(a, s_0) := \frac{s_0}{a^2} - s_0. \qquad (5)$$

For clarity, we refer to decreases in the spatial dimensions of an image as *dilation* and increases in the blurriness of an image as *scaling*. For a generalization to anisotropic scaling we replace scalar scale $t$ with matrix $T$, zero-scale $s_0$ with covariance matrix $\Sigma_0$, and dilation parameter $a$ with matrix $A$, so

$$T(A, \Sigma_0) = A^{-1}\Sigma_0 A^{-T} - \Sigma_0. \qquad (6)$$

**Semigroups** The semigroup property of Equation 3 is the gateway between classical scale-spaces and the group convolution [Cohen and Welling, 2016a] (see end of section). Semigroups $(S, \circ)$ consist of a non-empty set $S$ and a (binary) composition operator $\circ : S \times S \to S$. Typically, the composition $s \circ t$ of elements $s, t \in S$ is abbreviated $st$. For our purposes, these individual elements will represent dilation parameters. For $S$ to be a semigroup, it must satisfy the following two properties

- Closure: $st \in S$ for all $s, t \in S$
- Associativity: $(st)r = s(tr) = str$ for all $s, t, r \in S$

Note that commutativity is not a given, so $st \neq ts$. The family of Gaussian densities under spatial convolution is a semigroup[2]. Semigroups are a generalization of groups, which are used in Cohen and Welling [2016a] to model invertible transformations. For a semigroup to be a group, it must also satisfy the following conditions

- Identity element: there exists an $e \in S$ such that $es = se = s$ for all $s \in S$
- Inverses: for each $s \in S$ there exists a $s^{-1}$ such that $s^{-1}s = ss^{-1} = e$.

**Actions** Semigroups are useful because we can use them to model transformations, also known as (semigroup) *actions*. Given a semigroup $S$, with elements $s \in S$ and a domain $X$, the action $L_s^X : X \to X$ is a map, also written $x \mapsto L_s^X[x]$ for $x \in X$. The defining property of the semigroup action is that it is associative and closed under composition, inheriting its compositional structure from the semigroup $S$. There are, in fact, two versions of the action, a *left* and a *right* action (Equation 7). For the left action $L_{st}^X$, we first apply $L_t^X$ then $L_s^X$, but for the right action $R_{st}^X$, this is reversed.

$$\text{Left action: } L_{st}^X[x] = L_s^X[L_t^X[x]] \qquad \text{Right action: } R_{st}^X[x] = R_t^X[R_s^X[x]]. \qquad (7)$$

Actions can also be applied to functions $f : X \to Y$ by viewing $f$ as a point in a function space $F(X)$. Note, when the function domain is obvious, we just write $F$ for brevity. The result of the action is a new function, denoted $L_s^F[f]$. Since the domain of $f$ is $X$, we commonly write $L_s^F[f](x)$. Say the domain is a semigroup $S$ then an example left action is $L_s^F[f](x) = f(R_s^X[x]) = f(xs)$. This highlights a connection between left actions on functions and right actions on their domains, which we return to later in our discussion on nonlinearities. Another example, which we shall use later, is the action $S_{A,z}^{\Sigma_0}$ used to form scale-spaces, namely

$$S_{A,z}^{\Sigma_0}[f_0](x) = [G_A^{\Sigma_0} *_{\mathbb{Z}^d} f_0](A^{-1}x + z), \qquad G_A^{\Sigma_0} := G(\cdot, A^{-1}\Sigma_0 A^{-\top} - \Sigma_0). \qquad (8)$$

The elements of the semigroup are the tuples $(A, z)$ (dilation $A$, shift $z$) and $G_A^{\Sigma_0}$ is an anisotropic discrete Gaussian. The action first bandlimits by $G_A^{\Sigma_0} *_{\mathbb{Z}^d} f_0$ then dilates and shifts the domain by $A^{-1}x + z$. Note for fixed $(A, z)$ this maps functions on $\mathbb{Z}^d$ to functions on $\mathbb{Z}^d$.

**Lifting**   We can also view actions as maps from functions on $X$ to functions on the semigroup $S$, so $F(X) \to F(S)$. We call this a *lift* [Kondor and Trivedi, 2018], denoting lifted functions as $f^\uparrow$. One example is the scale-space action (Equation 8). If we set $x = 0$, then

$$f^\uparrow(A, z) = S_{A,z}^{\Sigma_0}[f_0](0) = [G_A^{\Sigma_0} *_{\mathbb{R}^d} f_0](z), \qquad (9)$$

which is the expression for an anisotropic scale-space (parameterized by the dilation $A$ rather than the scale $T(A, \Sigma_0) = A^{-1}\Sigma_0 A^{-T} - \Sigma_0$). To lift a function on to a semigroup, we do not necessarily have to set $x$ equal to a constant, but we could also integrate it out. An important property of *lifted* functions is that actions become simpler. For instance, if we define $f^\uparrow(s) = L_s^F[f](0)$ then

$$(L_t^F[f])^\uparrow(s) = L_s^F[L_t[f]](0) = L_{st}^F[f](0) = f^\uparrow(st). \qquad (10)$$

The action on $f$ could be complicated, like a Gaussian blur, but the action on $f^\uparrow$ is simply a 'shift' on $f^\uparrow(s) \mapsto f^\uparrow(st)$. We can then define the action $L_t^{F(S)}$ on lifted functions as $L_t^{F(S)}[f^\uparrow](s) = f^\uparrow(st)$. This action is another example of where the left action on the function is a right action on the domain.

**Equivariance and Group Correlations**   CNNs rely heavily on the (cross-)correlation[3]. Correlations $\star_{\mathbb{Z}^d}$ are a special class of linear map of the form

$$[f \star_{\mathbb{Z}^d} \psi](s) = \sum_{x \in \mathbb{Z}^d} f(x)\psi(x - s). \qquad (11)$$

Given a signal $f$ and filter $\psi$, we interpret the correlation as the collection of inner products of $f$ with all $s$-translated versions of $\psi$. This basic correlation has been extended to transformations other than translation via the *group correlation*, which as presented in Cohen and Welling [2016a] is

$$[f \star_H \psi](s) = \sum_{x \in X} f(x)\psi(L_{s^{-1}}^X[x]), \qquad (12)$$

where $H$ is the relevant group and $L_s^X$ is a *group* action e.g. for 2D rotation $L_s^X[x] = R_s x$, where $R_s$ is a rotation matrix. Most importantly, the domain of the output is $H$. It highlights how this is an inner product of $f$ and $\psi$ under all $s$-transformations of the filter. If we denote[4] $L_s^F[f](x) = f(L_{s^{-1}}^X[x])$, the correlation exhibits a special property. It is *equivariant* under actions of the group $H$. In math

$$L_s^{F(S)}[f \star_H \psi] = L_s^{F(X)}[f] \star_H \psi. \qquad (13)$$

Group correlation followed by the action is equivalent to the action followed by the group correlation, albeit the action is over a different domain. Note, the group action may 'look' different depending on whether it was applied before or after the group correlation, but it *represents* the exact same transformation.

**Notation**   As a shorthand, we just write $L_s$ from now on; the domain of the action should be obvious from context.

# 3 Method

We aim to construct a scale-equivariant convolution. We shall achieve this here by introducing an extension of the correlation to semigroups, which we then tailor to scalings.

**Semigroup Correlation** There are multiple candidates for a semigroup correlation $\star_S$. The basic ingredients of such a correlation will be the inner product, the semigroup action $L_s$, and the functions $\psi \in F$ and $f \in F$. Furthermore, it must be equivariant to (left) actions on $f$. For a semigroup $S$, domain $X$, and action $L_s : F \to F$, we define:

$$[\psi \star_S f](s) = \sum_{x \in X} \psi(x) L_s[f](x). \tag{14}$$

It is the set of responses formed from taking the inner product between a filter $\psi$ and a signal $f$ under all transformations of the signal. Notice that we transform the signal and not the filter and that we write $\psi \star_S f$, not $f \star_S \psi$—it turns out that a similar expression where we apply the action to the filter is not equivariant to actions on the signal. Furthermore this expression *lifts* a function from $X$ to $S$, so we expect actions on $f$ to look like a 'shift' on the semigroup. A proof of equivariance to (left) actions on $f$ is as follows

$$[\psi \star L_t[f]](s) = \sum_{x \in X} \psi(x) L_s[L_t[f]](x) = \sum_{x \in X} \psi(x) L_{st}[f](x) = [\psi \star f](st) = L_t[\psi \star f](s) \tag{15}$$

We have used the definition of the left action $L_s L_t = L_{st}$, the semigroup correlation, and our definition of the action for lifted functions. We can recover the standard 'convolution', by substituting $S = \mathbb{Z}^d$, $X = \mathbb{Z}^d$, and the translation action $L_s[f](x) = f(x + s)$:

$$[\psi \star_{\mathbb{Z}^d} f](s) = \sum_{x \in \mathbb{Z}^d} \psi(x) f(x + s) = \sum_{x' \in \mathbb{Z}^d} \psi(x' - s) f(x'). \tag{16}$$

where $x' = x + s$. We can also recover the group correlation by setting $S = H$, $X = H$, where $H$ is a discrete group, and $L_s[f](x) = f(R_s x)$, where $R_s$ is a right action acting on the domain $X = H$:

$$[\psi \star_H f](s) = \sum_{x \in H} \psi(x) f(R_s[x]) = \sum_{x' \in H} \psi(R_{s^{-1}}[x']) f(x'). \tag{17}$$

where $x' = R_s[x]$ and since $R_s$ is a *group* action inverses exist, so $x = R_{s^{-1}}[x']$. The semigroup correlation has a two notable differences from the group correlation: i) In the semigroup correlation we transform the signal and not the filter. When we restrict to the group and standard convolution, transforming the signal or the filter are equivalent operations since we can apply a change of variables. This is not possible in the semigroup case, since this change of variables requires an inverse, which we do not necessarily have. ii) In the semigroup correlation, we apply an action to the *whole* signal as $L_s[f]$, as opposed to just the domain ($f(R_s[x])$). This allows for more general transformations than allowed by the group correlation of Cohen and Welling [2016a], since transformations of the form $f(R_s[x])$ can only move pixel locations, but transformations of the form $L_s[f]$ can alter the values of the pixels as well, and can incorporate neighbourhood information into the transformation.

**The Scale-space Correlation** We now have the tools to create a scale-equivariant correlation. All we have to choose is an appropriate action for $L_s$. We choose the scale-space action of Equation 8. The scale-space action $S_{z,A}^{\Sigma_0}$ for functions on $\mathbb{Z}^d$ is given by

$$S_{A,z}^{\Sigma_0}[f](x) = [G_A^{\Sigma_0} \star_{\mathbb{Z}^d} f](A^{-1}x + z), \qquad S_{A,z}^{\Sigma_0} S_{B,y}^{\Sigma_0} = S_{AB,Ay+z}^{\Sigma_0}. \tag{18}$$

Since our scale-space correlation only works for discrete semigroups, we have to find a suitable discretization of the dilation parameter $A$. Later on we will choose a discretization of the form $A_k = 2^{-k}I$ for $k \geq 0$, but for now we will just assume that there exists some countable set $\mathcal{A}$, such that $S = \{(A, z)_{A \in \mathcal{A}, z \in \mathbb{Z}^d}\}$ is a valid discrete semigroup. We begin by assuming we have lifted an input image $f$ on to the scale-space via Equation 9. The lifted signal is indexed by coordinates $A, z$ and so the filters share this domain and are of the form $\psi(A, z)$. The scale-space correlation is then

$$[\psi \star_S f](A, z) = \sum_{(B,y) \in S} \psi(B, y) S_{B,y}^{\Sigma_0}[f](A, y) = \sum_{B \in \mathcal{A}} \sum_{y \in \mathbb{Z}^d} \psi(B, y) f(BA, A^{-1}y + z). \tag{19}$$

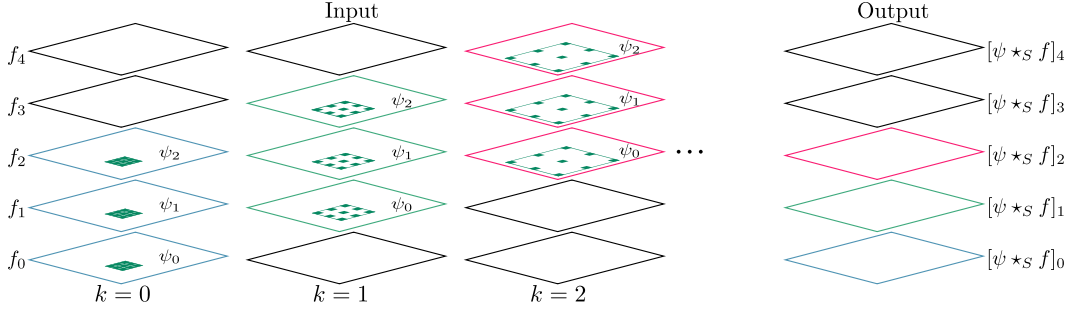

Figure 3: Scale correlation schematic: The left 3 stacks are the same input $f$, with levels $f_\ell(\cdot) = f(2^{-\ell}I, \cdot)$. Each stack shows the inner product between filter $\psi$ (in green) at translation $z$ for dilation $2^{-k}I$ corresponding to the output level $[\psi \star_S f]_k$ on the right with matching color. Notice that as we dilate the filter, we also shift it one level up in the scale-space, accoridng to Equation 20.

For the second equality we recall the action on a lifted signal is governed by Equation 10. The appealing aspect of this correlation is that we do not need to convolve with a bandlimiting filter—a potentially expensive operation to perform at every layer of a CNN—since we use signals that have been lifted on to the semigroup. Instead, the action of scaling by $A$ is accomplished by 'fetching' a slice $f(BA, \cdot)$ from a blurrier level of the scale-space. Let's restrict the scale correlation to the scale-space where $A_k = 2^{-k}I$ for $k \geq 0$, with zero-scale $\Sigma_0 = \frac{1}{4}I$. Denoting $f(2^{-k}I, \cdot)$ as $f_k(\cdot)$, this can be seen as a dilated convolution [Yu and Koltun, 2015] between $\psi_\ell$ and slice $f_{\ell+k}$. This form of the scale-space correlation (shown below) we use in our experiments. A diagram of this can be seen in Figure 3:

$$[\psi \star_S f]_k(z) = \sum_{\ell \geq 0} \sum_{y \in \mathbb{Z}^d} \psi_\ell(y) f_{\ell+k}(2^k y + z). \qquad (20)$$

**Equivariant Nonlinearities**   Not all nonlinearities commute with semigroup actions, but it turns out that pointwise nonlinearities $\nu$ commute with a special subset of actions of the form,

$$L_s[f](x) = f(R_s[x]) \qquad (21)$$

where $R_s$ is a right action on the domain of $f$. For these sorts of actions, we cannot alter the values of the $f$, just the locations of the values. If we write function composition as $[\nu \bullet f](x) = \nu(f(x))$, then a proof of equivariance is as follows:

$$[\nu \bullet L_s[f]](x) = \nu(f(R_s[x])) = [\nu \bullet f](R_s[x]) = L_s[\nu \bullet f](x). \qquad (22)$$

Equation 21 may at first glance seem overly restrictive, but it turns out that this is not the case. Recall that for functions lifted on to the semigroup, the action is $L_s[f](x) = f(xs)$. This satisfies Equation 21, and so we are free to use pointwise nonlinearities.

**Batch normalization**   For batch normalization, we compute batch statistics over all dimensions of an activation tensor except its channels, as in Cohen and Welling [2016a].

**Initialization**   Since our correlations are based on dilated convolutions, we use the initialization scheme presented in [Yu and Koltun, 2015]. For pairs of input and output channels the center pixel of each filter is set to one and the rest are filled with random noise of standard deviation $10^{-2}$.

**Boundary conditions**   In our semigroup correlation, the scale dimension is infinite—this is a problem for practical implementations. Our solution is to truncate the scale-space to finite scale. This breaks global equivariance to actions in $S$, but is locally correct. Boundary effects occur at activations with receptive fields covering the truncation boundary. To mitigate these effects we: i) use filters with a scale-dimension no larger than two, ii) interleave filters with scale-dimension 2 with filters of scale-dimension 1. The scale-dimension 2 filters enable multiscale interactions but they propagate boundary effects; whereas, the scale-dimension 1 kernels have no boundary overlap, but also no multiscale behavior. Interleaving trades off network expressiveness against boundary effects.

Table 1: Results on the Patch Camelyon and Cityscapes Dataset. Higher is better. Our scale-equivariant model outperforms the matched baselines. We must caution that better competing results can be found in the literature, when the computational constraint is relaxed. For instance, Shelhamer et al. [2019] report a mAP of 71.4 on a ResNet-34, which is deeper than our model by 15 layers.

| PCam Model | Accuracy |
|---|---|
| DenseNet Baseline | 87.0 |
| S-DenseNet (Ours) | 88.1 |
| [Veeling et al., 2018] | 89.8 |

| Cityscapes Model | mAP |
|---|---|
| ResNet, matched parameters | 45.66 |
| ResNet, matched channels | 49.99 |
| S-ResNet, multiscale (Ours) | 63.53 |
| S-ResNet, no interaction (Ours) | 64.78 |

**Scale-space implementation**  We use a 4 layer scale-space and zero-scale $1/4$ with dilations at integer powers of 2, the maximum dilation is 8 and kernel width 33 (4 std. of a discrete Gaussian). We use the discrete Gaussian of Lindeberg [1990]. In 1D for scale parameter $t$, this is

$$G(x, t) = e^{-t} I_{|x|}(t) \tag{23}$$

where $I_x(t)$ is the modified Bessel function of integer order. For speed, we make use of the separability of isotropic kernels. For instance, convolution with a 2D Gaussian can we written as a convolution with 2 identical 1D Gaussians sequentially along the $x$ and then the $y$-axis. For an $N \times N$ input image and $M \times M$ blurring kernel, this reduces computational complexity of the convolution as $O(M^2 N^2) \to O(2MN^2)$. With GPU parallelization, this saving is $O(M^2) \to O(M)$, which is especially significant for us since the largest blur kernel we use has $M = 33$.

**Multi-channel features**  Typically CNNs use multiple channels per activation tensor, which we have not included in our above treatment. In our experiment we include input channels $i$ and output channels $o$, so a correlation layer is

$$[\psi \star_S f]_k^o(z) = \sum_i \sum_{\ell \geq 0} \sum_{x \in \mathbb{Z}^d} \psi_\ell^{i,o}(x) f_{\ell+k}^i(2^k x + z). \tag{24}$$

## 4   Experiments and Results

Here we present results for some preliminary simple experiments on the Patch Camelyon [Veeling et al., 2018] and Cityscapes [Cordts et al., 2016] datasets. Due to the extra computational overhead of computing the scale-equivariance cross-correlation, the experiments are more an indicator of what our method is capable of, but please note that the baselines are non state-of-the-art reimplementations, restricted in size for fair-comparison rather than to prove we are beating any benchlines. We also visualize the quality of scale-equivariance achieved.

**Patch Camelyon**  The Patch Camelyon or PCam dataset [Veeling et al., 2018] contains 327 680 tiles from two classes, metastatic (tumorous) and non-metastatic tissue. Each tile is a $96 \times 96$ px RGB-crop labelled as metastatic if there is at least one pixel of metastatic tissue in the central $32 \times 32$ px region of the tile. We test a 4-scale DenseNet model [Huang et al., 2017], 'S-DenseNet', on this task (architecture in supplement). We also train a scale non-equivariant DenseNet baseline and the rotation equivariant model of Veeling et al. [2018]. Our training procedure is: 100 epochs SGD, learning rate 0.1 divided by 10 every 40 epochs, momentum 0.9, batch size of 512, split over 4 GPUs. For data augmentation, we follow the procedure of Veeling et al. [2018], Liu et al. [2017], using random flips, rotation, and 8 px jitter. For color perturbations we use: brightness delta 64/255, saturation delta 0.25, hue delta 0.04, constrast delta 0.75. The evaluation metric we test on is accuracy. The results in Table 1 show that both the scale and rotation equivariant models outperform the computation-matched baseline.

**Cityscapes**  The Cityscapes dataset [Cordts et al., 2016] contains 2975 training images, 500 validation images, and 1525 test images of resolution $2048 \times 1024$ px. The task is semantic segmentation into 19 classes. We train a 4-scale ResNet He et al. [2016], 'S-ResNet', and baseline. We train an equivariant network with and without multiscale interaction layers. We also train two scale non-equivariant models, one with the same number of channels, one with the same number of parameters. Our training procedure is: 100 epochs Adam, learning rate $10^{-3}$ divided by 10 every 40 epochs, batch

size 8, split over 4 GPUs. The results are in Table 1. The evaluation metric is mean average precision. We see that our scale-equivariant model outperforms the baselines. We must caution however, that better competing results can be found in the literature. For instance, Shelhamer et al. [2019] report a mAP of 71.4 on a ResNet-34, which is deeper than our model by 15 layers. That said they train with continuous scale augmentation, which we do not. The reason our baseline underperforms compared to the literature is because of the parameter/channel-matching, which have shrunk its size somewhat due to our own resource constraints. On a like-for-like comparison scale-equivariance appears to help.

**Quality of Equivariance** We validate the quality of equivariance empirically by comparing activations of a dilated image against the theoretical action on the activations. Using $\Phi$ to denote the deep scale-space (DSS) mapping, we compute the normalized $L2$-distance at each level $k$ of a DSS. Mathematically this is

$$L(2^{-\ell}, k) = \frac{\|\Phi[f](k+\ell, 2^{\ell}\cdot) - \Phi[S_{2^{-\ell},0}^{1/4I}[f]](k,\cdot)\|_2}{\|\Phi[f](k+\ell, 2^{\ell}\cdot)\|_2}. \tag{25}$$

The equivariance errors are in Figure 4 for 3 DSSs with random weights and a scale-space trucated to 8 scales. We see that the average error is below 0.01, indicating that the network is mostly equivariant, with errors due to truncation of the discrete Gaussian kernels used to lift the input to scale-space. We also see that the equivariance errors blow up for constant $\ell + k$ in each graph. This is the point where the receptive field of an activation overlaps with the scale-space truncation boundary.

## 5 Related Work

In recent years, there have been a number of works on group convolutions, namely continuous roto-translation in 2D [Worrall et al., 2017] and 3D [Weiler et al., 2018a, Kondor et al., 2018, Thomas et al., 2018] and discrete roto-translations in 2D [Cohen and Welling, 2016a, Weiler et al., 2018b, Bekkers et al., 2018, Hoogeboom et al., 2018] and 3D [Worrall et al., 2017], continuous rotations on the sphere [Esteves et al., 2018, Cohen et al., 2018b], in-plane reflections [Cohen and Welling, 2016a], and even reverse-conjugate symmetry in DNA sequencing [Lunter and Brown, 2018]. Theory for convolutions on compact groups—used to model invertible transformations—also exists [Cohen and Welling, 2016a,b, Kondor and Trivedi, 2018, Cohen et al., 2018a], but to date the vast majority of work has focused on rotations.

For scale, there far fewer works with explicit scale equivariance. Henriques and Vedaldi [2017], Esteves et al. [2017] both perform a log-polar transform of the signal before passing to a standard CNN. Log-polar transforms reparameterize the input plane into angle and log-distance from a predefined origin. The transform is sensitive to origin positioning, which if done poorly breaks translational equivariance. Marcos et al. [2018] use a group CNN architecture designed for roto-translation, but instead of rotating filters in the group correlation, they scale them. This seems to work on small tasks, but ignores large scale variations. Kanazawa et al. [2014] convolve the same filter over rescaled versions of the same feature map and then max-pool over feature location, for local scale-invariance. Hilbert et al. [2018] instead use filters of different sizes, but without any attention to equivariance. Ke et al. [2017] introduce a multigrid convolution, where the convolution outputs multiple feature maps at different resolutions. The input to each resolution convolution is a concatenation of rescaled feature maps from the previous layer. This is similar to our work, but differing in two ways, 1) there is no across-scale weight-tying (so no explicit equivariance), and 2)

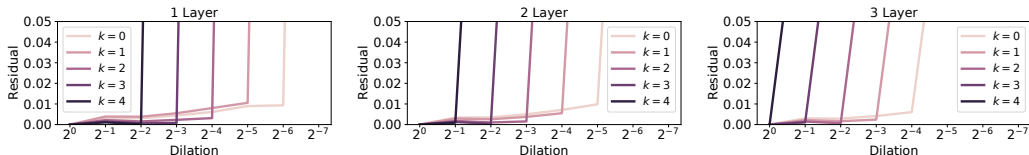

Figure 4: Equivariance quality. Left to right: 1, 2, and 3 layer DSSs. Each line represents the error as in Equation 25. We see the residual error is typically $< 0.01$ until boundary effects are present.

they maintain feature maps at difference resolutions rather than different scalings (bandlimits). Two other works, with similar approaches are Huang et al. [2018], Saxena and Verbeek [2016], but their goals are not scale equivariance, but architecture search. Feature Pyramid Networks [Lin et al., 2017] also uses multiscale features, using a scheme similar to a UNet [Ronneberger et al., 2015], but where predictions are made using every resolution of decoded feature maps. An interesting work with a very different approach is Shelhamer et al. [2019], where filters are formed as the convolution of a base filter and an anisotropic Gaussian filter, whose covariance is predicted at test time. This produces scale-adaptive filters.

# 6 Discussion, Limitations, and Future Works

We found our best performing architectures were composed mainly of correlations where the filters' scale dimension is one, interleaved with correlations where the scale dimension is higher. This is similar to a network-in-network [Lin et al., 2013] architecture, where $3 \times 3$ convolutional layers are interleaved with $1 \times 1$ convolutions. We posit this is the case because of boundary effects, as were observed in Figure 4. Further to boundary effects, we also suspect that using non-integer dilations with finer increments in scale would improve performance greatly, as often witnessed in the scale-space literature. This would, however, involve the development of non-integer dilations and hence interpolation. We see working on mitigating boundary effects, and using a semigroup correlation for non-integer scalings as an important future work, not just for scale-equivariance, but CNNs as a whole.

Another limitation of the current model is the increase in computational overhead, since we have added an extra dimension to the activations. This may not be a problem long term, as GPUs grow in speed and memory, but the computational complexity of a correlation grows exponentially in the number of symmetries of the model, and so we need more efficient methods to perform correlation, either exactly or approximately. In terms of experimentation, this extra computation limited our ability to compare against large state-of-the-art models, perhaps giving our model an unfair advantage in this limited scheme.

In terms of the experiments, a much more in depth exploration of the empirical properties of the scale-equivariant correlation is needed. Our light proof-of-concept experiments provide some evidence that built-in scale equivariance can help, but before this is going to be useful in practise, we need to solve the issues of efficiency, non-integer dilations, and the boundary effects due to scale-space truncation.

We see semigroup correlations as an exciting new family of operators to use in deep learning. We have demonstrated a proof-of-concept on scale, but there are many semigroup-structured transformations left to be explored, such as causal shifts, occlusions, and affine transformations. Concerning scale, we are also keen to explore how multi-scale interactions can be applied on other domains as meshes and graphs, where symmetry is less well-defined.

# 7 Conclusion

We have presented deep scale-spaces a generalization of convolutional neural networks, exploiting the scale symmetry structure of conventional image recognition tasks. We outlined new theory for a generalization of the convolution operator on semigroups, the semigroup correlation. Then we showed how to derive the standard convolution and the group convolution [Cohen and Welling, 2016a] from the semigroup correlation. We then tailored the semigroup correlation to the scale-translation action used in classical scale-space theory and demonstrated how to use this in modern neural architectures.

### Acknowledgements

We thanks Koninklijke Philips N.V. for in-cash and in-kind support of this research. We also thank Rianne van den Berg, Patrick Forré, and the anonymous reviewers who all made important contributions to this paper.

## Footnotes

[2]For the Gaussians we find the identity element as the limit $\lim_{t \downarrow 0} G(\cdot, t)$, which is a Dirac delta. Note that this element is not strictly in the set of Gaussians, thus the Gaussian family has no identity element.

[3]In the deep learning literature these are inconveniently referred to as convolutions, but we stick to correlation.

[4]Recall how earlier we gave an example of left actions on functions as $L_s^F[f](x) = f(R_s^X[x]) = f(xs)$, the group action we present here is an example of this, because $L_{s^{-1}}^X[x]$ is a right action.

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
