[Supplementary Material · supplement.pdf]

# Supplementary Material
# Deep Scale-spaces: Equivariance Over Scale

## Abstract

In this supplementary material we elaborate on where the scale-space action comes
from and the architectures used in our paper.

## 1 Scale-spaces

Here we provide some extra information on scale-spaces for the interested reader. For in depth
literature, we suggest Florack et al. [1994, 1992], Pauwels et al. [1995], Lindeberg [1990, 1997],
Crowley et al. [2002], Salden et al. [1998], Duits et al. [2004, 2003], Duits and Burgeth [2007],
Burgeth et al. [2005a,b].

### 1.1 1D Gaussian Scale-space

We are given an initial 1D signal $f_0$ with intrinsic bandlimit, or *zero-scale*, defined by $s_0$ i.e.,
we impose a maximum frequency, such that there is a correspondence with discretized signals.
Note that $\sqrt{s_0}$ is inversely proportional to frequency content of the signal. We wish to downsize
it isotropically by a factor $a$, which we call the *dilation*. For this we introduce the downsizing
action $L_a[f](x) = f(a^{-1}x)$ for $a \leq 1$. We model the bandlimit of the initial signal as the result of
convolving some other signal $f$ with a Gaussian of width $s_0$, so

$$f_0(x) = [G(\cdot, s_0) *_{\mathbb{R}^d} f](x). \tag{1}$$

Now the result of the downscaling action of $f_0$ is as follows

$$L_a[f_0](x) = f_0(a^{-1}x) = \int_{\mathbb{R}^d} G(a^{-1}x - y, s_0)f(y)\,\mathrm{d}y = \int_{\mathbb{R}^d} G(x - ay, a^2 s_0)f(y)\,a\mathrm{d}y \tag{2}$$

$$= \int_{\mathbb{R}^d} G(x - z, a^2 s_0)f(a^{-1}z)\,\mathrm{d}z = [G(\cdot, a^2 s_0) *_{\mathbb{R}^d} L_a[f]](x). \tag{3}$$

From the first to second lines we have performed a change of variables $z = ay$. So we see that the
effect of downsizing a bandlimited signal by $a$ shifts the bandlimit from $s_0$ to $a^2 s_0$. Since $a \leq 1$, this
means the blurring Gaussian is narrower and so the frequency content of the signal has been shifted
higher. The key relation to bear in mind is the shift $s_0 \mapsto a^2 s_0$.

For a proper scaling, we want the result of the downscaling to have the *same bandlimit* as the original
signal $f_0$. This is because if we are representing the signal on a discrete grid, then we have a physically
defined maximum frequency content we can store, given by the pixel separation. The solution is to
convolve the signal $L_a[f_0]$ with a correcting Gaussian of width $s_0 - a^2 s_0$. Note that this is possible,
since $a \leq 0$, so $s_0 - a^2 s_0 > 0$. Alternatively, we want to find a correcting Gaussian to blur before
downsizing. Say this correcting Gaussian has bandlimit $t$, then we have

$$L_a[G(\cdot, t) * f_0] = L_a[G(\cdot, t + s_0) * f] = [G(\cdot, a^2(t + s_0)) * L_a[f]]. \tag{4}$$

26　But we also want the bandlimit of the downscaled signal to be $s_0$, so we have the relation

$$a^2(t + s_0) = s_0. \tag{5}$$

27　Thus a 1D Gaussian scale-space, parameterized by dilation, can be built by setting

$$f(t(a, s_0), x) = [G(\cdot, t(a, s_0)) * f_0](x), \qquad t(a, s_0) := \frac{s_0}{a^2} - s_0 \tag{6}$$

$$f(0, x) = f_0(x) \tag{7}$$

## 28　1.2　$N$D Gaussian Scale-space

29　We now explore downscaling in $N$-dimensions. We first of all represent an initial image $f_0$ as

$$f_0 = G(\cdot, \Sigma_0) *_{\mathbb{R}^d} f. \tag{8}$$

30　We use the Gaussian to represent the fact that $f_0$ should have an intrinsic bandlimit, usually defined
31　by the resolution at which it is sampled. Now let's introduce the affine action:

$$L_{A,z}[f](x) = f(A^{-1}(x - z)). \tag{9}$$

32　It simply applies an affine transformation to our signal. Now using a similar logic to in the 1D case,
33　if we concatenate the affine action with bandlimiting we get

$$L_{A,z}[G(\cdot, \Sigma_0) *_{\mathbb{R}^d} f_0] = G(\cdot, A\Sigma_0 A^\top) *_{\mathbb{R}^d} L_{A,z}[f_0]. \tag{10}$$

34　So we see that resizing a bandlimited signal shifts the bandlimit according to $\Sigma_0 \mapsto A\Sigma_0 A^\top$. Since
35　we would like to have the same bandlimit on our signal before and after the resizing (since we can
36　only represent the signal at constant resolution), we introduce a second bandlimiting by convolving
37　with a Gaussian of width $\Sigma_0 - A\Sigma_0 A^\top$. To save space, we write $G_\Sigma = G(\cdot, \Sigma)$. So

$$G_{\Sigma_0} *_{\mathbb{R}^d} L_{A,z}[f] = G_{\Sigma_0 - A\Sigma_0 A^\top} *_{\mathbb{R}^d} L_{A,z}[f_0] = L_{A,z}[G_{A^{-1}\Sigma_0 A^{-\top} - \Sigma_0} *_{\mathbb{R}^d} f_0] \tag{11}$$

38　From the first to the second equality, we have exchanged the order of the Gaussian convolution and
39　the affine action and altered the bandwidth from $\Sigma_0 - A\Sigma_0 A^\top$ to $A^{-1}\Sigma_0 A^{-\top} - \Sigma_0$, which comes
40　from the relation established in Equation 10. Thus the affine action for an affine scale-pyramid is
41　defined as

$$L_{A,z}[G_{A^{-1}\Sigma_0 A^{-\top} - \Sigma_0} *_{\mathbb{R}^d} f](x) = [G_A^{\Sigma_0} *_{\mathbb{R}^d} f](A^{-1}(x - z)) \tag{12}$$

42　where we have defined $G_A^{\Sigma_0} = G_{A^{-1}\Sigma_0 A^{-\top} - \Sigma_0}$.

43　For what values of $A$ and $z$ is this action valid? Let's first focus on $A$. To maintain the zero-scale of
44　$\Sigma_0$, we had to convolve with a Gaussian of width $\Delta = \Sigma_0 - A\Sigma_0 A^\top$. Now we know that covariance
45　matrices have to be symmetric $\Delta = \Delta^\top$ and positive definite $\Delta \succ 0$. We see already that it is
46　symmetric, but it is not necessary positive definite. If the base bandlimit is of the form $\Sigma_0 = \sigma_0^2 I$
47　(the original image is isotropically bandlimited), then we can rearrange to

$$\Delta = \Sigma_0 - A\Sigma_0 A^\top = (I - AA^\top)\Sigma_0 \succ 0 \tag{13}$$

48　This expression is only positive definite if $I - AA^\top \succ 0$; that is

$$I \succ AA^\top. \tag{14}$$

49　This condition implies that $A^\top$ is a contraction because

$$v^\top(I - AA^\top)v = \|v\|_2^2 - \|A^\top z\|_2^2 \geq 0 \implies \|v\|_2^2 \quad \geq \|A^\top v\|_2^2, \qquad \forall v \in \mathbb{R}^N. \tag{15}$$

50　Another way of phrasing this is the that the singular values of $A$ may not exceed unity. Note that
51　rotations do not break this constraint. So we see this this model naturally aligns with our notion that
52　we can only model image downscalings, and that upscalings are prohibited.

## 1.3 Other Scale-space variants

We have presented the Gaussian scale-space in $N$D, but there also exists a zoo of other scales-spaces. The most prominent are: the $\alpha$-scale-spaces [Pauwels et al., 1995], the discrete Gaussian scale-spaces [Lindeberg, 1990], and the binomial scale-spaces [Burt, 1981]. In the following, we give a brief introduction to each, exhibited in 1D.

**$\alpha$-scale-spaces** $\alpha$-scale-spaces Pauwels et al. [1995] are a generalization of the Gaussian space-space in the continuous domain. They are easiest to understand by considering their form in Fourier-space. We begin by considering the Fourier transform of the Gaussian space-space over the spatial dimension

$$\hat{f}(t,\omega) = \hat{G}(\omega,t) \cdot \hat{f}_0(\omega) \tag{16}$$

$$\hat{f}(0,\omega) = \hat{f}_0(\omega), \tag{17}$$

where $\hat{f}$ is the Fourier transform of $f$. We are interested in finding a collection of filters like $G$, closed under convolution. In the Fourier domain this corresponds to finding a collection of filters, like $\hat{G}$ closed under multiplication. The Fourier transform of the Gauss-Weierstrass kernel is

$$G(x,t) = \frac{1}{(4\pi t)^{1/2}} \exp\left\{-\frac{x^2}{4t}\right\} \stackrel{\text{FT}}{\Longleftrightarrow} \hat{G}(\omega,t) = \exp\left\{-\omega^2 t\right\}. \tag{18}$$

The collection $\{\hat{G}(\omega,t)\}_{t>0}$ is indeed closed under multiplication and forms a semigroup. To form the $\alpha$-scale-spaces we notice that the Fourier kernel

$$\hat{G}^\alpha(\omega,t) = \exp\left\{-\omega^{2\alpha} t\right\} \tag{19}$$

is also closed under multiplication and defines a semigroup. The range of $\alpha$ is typically taken to be $(0,1]$, to make sure that higher levels in the $\alpha$-scale-space are blurrier. Notice that for $\alpha = 1$ we return to the standard Gaussian scale-space.

**Binomial Scale-space** The binomial scale-space Crowley et al. [2002] is a discrete scale-space in both the spatial and scale dimensions. It is generated by convolution in $\mathbb{Z}$ with the binomial kernel

$$B(x,N) = {}^N C_x \left/ \sum_{x=0}^{N} {}^N C_x \right., \qquad {}^N C_x = \frac{N!}{(N-x)!N!}, \tag{20}$$

where $N > 0$ is the width of the kernel and $0 \leq x \leq N$ is the spatial location of the filter tap. Thus the scale-space is

$$f(N,x) = [B(\cdot,N) *_{\mathbb{Z}} f_0](x) \qquad N > 0. \tag{21}$$

As $N$ grows $B(N,x)$ rapidly converges to a Gaussian kernel of variance $\sigma^2 = N/4$. The Binomial filters are closed under convolution obeying the semigroup property

$$[B(\cdot,N) *_{\mathbb{Z}} B(\cdot,M)](x) = B(x, N+M-1). \tag{22}$$

**Discrete Gaussian Scale-space** The discrete Gaussian scale-space Lindeberg [1990] is discrete in the spatial dimension but continuous in the scale dimension, which makes it popular to work with in many practical scale-spaces with non-integer dilation. The scale-space is generated by convolution in $\mathbb{Z}$ with the discrete Gaussian kernel

$$G(x,t) = e^{-t} I_{|x|}(t), \qquad I_x(t) = \sum_{m=0}^{\infty} \frac{\left(\frac{1}{2}x\right)^{2m+\alpha}}{m!\Gamma(m+\alpha+1)} \tag{23}$$

where the term $I_k(t)$ is a modified Bessel function of the first kind. These can be implemented easily using `scipy.special.ive`. The scale-space is formed in the usual way as

$$f(t,x) = [G(\cdot,t) *_{\mathbb{Z}} f_0](x) \qquad t > 0 \tag{24}$$

$$f(0,x) = f_0(x). \tag{25}$$

Table 1: A residual network. Input at the top. A horizontal line denotes spatial average pooling of stride 2, kernel size 2. Shape is displayed as [scale-space levels, height, width, channels out]. The no-res block denotes a residual block without the skip connection, i.e. $y = \mathcal{F}(x)$. Scale pooling denotes an averaging over all scale dimensions.

| Layer type | Shape |
|---|---|
| res$[k, 3, 3]$ | $[S, 992, 992, N]$ |
| res$[k, 3, 3]$ | $[S, 496, 496, 2N]$ |
| res$[1, 3, 3]$ | $[S, 248, 248, 4N]$ |
| res$[k, 3, 3]$ | $[S, 248, 248, 4N]$ |
| res$[1, 3, 3]$ | $[S, 124, 124, 8N]$ |
| res$[1, 3, 3]$ | $[S, 124, 124, 8N]$ |
| res$[1, 3, 3]$ | $[S, 124, 124, 8N]$ |
| res$[k, 3, 3]$ | $[S, 124, 124, 8N]$ |
| no-res$[k, 3, 3]$ | $[S, 124, 124, 8N]$ |
| scale-pool | $[1, 124, 124, 8N]$ |
| corr[1,1,1], | $[1, 124, 124, 19]$ |
| bilinear upsample | $[1, 992, 992, 19]$ |

## 2  Architectures In The Experiments

In the experiments, we use a DenseNet Huang et al. [2017] and a ResNet He et al. [2016]. The architectures are as follows. For the scale equivariant versions, we use 4 scales of a discrete Gaussian scale-space Lindeberg [1990].

**ResNet** The residual network consists of a concatenation of residual blocks. A single residual block implements the following

$$y = x + \mathcal{F}(x) \tag{26}$$

where on the RHS we refer to $x$ as the skip connection and $\mathcal{F}(x)$ as the residual connection. If $x$ has fewer channels than $\mathcal{F}(x)$, then we pad the missing dimensions with zeros. Each residual connection uses a concatenation of two scale-equivariant correlation interleaved with batch normalization (BN) and a ReLU (ReLU) nonlinearity. These are composed as follows (input left, output right)

$$\text{corr}[1, 3, 3] \text{ - BN - ReLU - corr}[k, 3, 3] \text{ - BN.} \tag{27}$$

where corr$[k, h, w]$ refers to a scale correlation with kernel size $[k, h, w]$ and where $k$ is the number of scale channels, $h$ is the spatial height of the filter, and $w$ is its spatial width. We denote the entire residual block as res$[k, h, w]$.

The model we use is given in Table 1. It follows the practice of Yu et al. [2017], who use a bilinear upsampling at the end of the network, since segmentations do not tend to contain high frequency details. In our experiments we use the models shown in Table 2

**DenseNet** The Dense network Huang et al. [2017] consists of a concatenation of 3 dense blocks. Each dense block is composed of layers of the form

$$y_{N+1} = \mathcal{H}\left([y_1, y_2, ..., y_N]\right) \tag{28}$$

where $[y_1, y_2, ..., y_N]$ is the concatenation of all the previous layers' outputs. Each layer $\mathcal{H}$ is the composition (input left, output right)

$$\text{BN - ReLU - corr}[k, 3, 3] \tag{29}$$

Table 2: We match model settings with their names from the paper. Settings are displayed as $[k, S, N]$ or [kernels scale dim., num scales, number of channels].

| Model | Settings |
|---|---|
| S-ResNet, multiscale interaction | $[2, 4, 16]$ |
| S-ResNet no interaction | $[1, 4, 16]$ |
| ResNet, matched channels | $[1, 1, 16]$ |
| ResNet, matched parameters | $[1, 1, 18]$ |

Table 3: A DenseNet. Input at the top. For shape, we show the number of scales, the height, the width, and the number of channels. $S$ denotes the number of scales used per layer.

| Layer type | Shape |
|---|---|
| dense$_{12}$[1, 3, 3] ×3 | [S, 96, 96, 39] |
| transition[3, 3, 3] | [S, 48, 48, 19] |
| dense$_{24}$[1, 3, 3] ×3 | [S, 48, 48, 94] |
| transition[3, 3, 3] | [S, 24, 24, 47] |
| dense$_{48}$[1, 3, 3] ×3 | [S, 24, 24, 213] |
| Global average pooling | [1, 1, 213] |
| Linear layer | [1, 1, 2] |

where corr$[k, 3, 3]$ was described in the previous section. We use the notation dense$_C[k, h, w] \times N$ to denote a dense block with $N$ layers and $C$ output channels per layer. The number of channel outputs remains constant per layer within a dense block. Between dense blocks, we insert transition layer which have the form

$$\text{dense}[1, 1, 1] \times 1 \text{ - pool - dense}[k, h, w] \times 1. \tag{30}$$

Here we use a $1 \times 1$ convolution to halve the number of output channels, and then perform a spatial average pooling with kernel size 2 and stride 2, followed by a second dense layer. We denote these as transition$[k, h, w]$. We also use long skip connection between transistion layers. The network we use is shown in Table 3.