[Reviews · NeurIPS 2019]

Reviewer 1



POST-REBUTTAL COMMENTS Notation: Thanks for considering my remarks. Your proposed modifications sound good. Quality of results: I mostly found the theoretical framework and the subject of scale-equivariance to be interesting, therefore I did not require the paper to achieve state-of-the-art results. Related work: Thanks for following the suggestions of the other reviewers. The FPN paper is particularly relevant because I believe that they apply the same output function to all levels of the representation, encouraging some degree of scale-equivairance. My overall rating of the paper remains unchanged. Other comments... Related work: Another interesting multi-scale (but not scale-equivariant) paper is "convolutional neural fabrics". Aside: After reviewing this paper, a question came back to bug me... Isn't normal convolution on non-infinite signals a kind of semi-group convolution? That is, if we take the domain to be the non-negative integers {0, 1, 2, ...} and the operation to be addition, this forms a semi-group (closed and associative but without an inverse). However, we deal with finite signals all the time (by inserting zeros or restricting the output i.e. "same" or "valid" padding). Could a simple approach such as this work here? I have probably missed something because this doesn't incorporate the notion of band-limiting. Anyway, it might be good if you could address this somehow. ORIGINAL REVIEW # Originality This paper develops an elegant and rigorous framework for scale-equivariant conv-nets. Much of the originality lies in the development of this theory, and this is definitely sufficient for me. It might be good to cite: - Kanazawa et al. ""Locally Scale-Invariant Convolutional Neural Networks" (NIPS Workshop 2014) # Quality Overall, the quality of the theoretical investigation and the experimental procedure is very high. I wonder whether the inductive bias of scale-equivariance provides better performance for the same amount of training data. I would have like to see a graph showing accuracy versus amount of training data for scale-equivariant and normal CNNs. In equation 22, I don't think there is any need for L'_s[x] to be a left-action? Since the notation "L_" has been used for left-actions, maybe it's better to use a different notation for this function of s and x? Furthermore, the map L'_s[x] = xs is actually a right-action not a left-action, since L'_s[L'_t[x]] = L'_s[xt] = xts = L'_ts[x]. It is stated that one advantage of semi-group convolution is that we can apply an action to the whole signal rather than just a transformation of its input domain. However, all of the examples use L_s[f](x) = f(x s). Can you provide one example of a semi-group convolution that goes beyond this restriction? (I realise that the restriction is later required for pointwise non-linearities.) Was any pre-training used for the segmentation task? I wonder whether even larger gains could be expected if the scale-equivariant architecture were employed during pre-training? # Clarity The writing is clear and the approach is well motivated. However, I found the use of the symbol L_s to be really confusing. Please see that additional section below. The experimental details are perfectly clear. This makes the paper highly reproducible. The initial introduction of s_0 was slightly confusing in the text. It might help to move equation 1 from the supplementary material to the main paper? In this case, you might want to use a different symbol for f, since it is overloaded. I mostly followed Section 1.2 in the supplementary material except: - I thought it might be more clear to use f rather than f_0 in equation 10. (Technically both are correct since the condition holds for all f.) - I did not understand how the middle expression in equation 11 was obtained, although I agree with the equality of the first and last terms. Could you clarify the explanation? # Significance The tool of semi-group convolution and the development of scale-equivariant conv-nets may both have a significant impact on the field. # Notation Here is a description of the confusion that I encountered as I read the paper: The first difficulty came at line 112. I did not realise immediately that there was no relationship between this L_s and the L_s from three lines earlier. Furthermore, when you say X = S, as a reader my mind immediately goes to the group action L_s[x] = s x, and it is confusing to see L_s[f](x) = f(x s) with x and s in the opposite order. It is correct, but it would be nice to walk the reader through this more gently. The next confusion came at line 129. In equation 12, "L_s is a group action" defined on the group H. However, in equation 13, it seems that L_s denotes an action defined on functions on H. Should we assume in general that, when L_s[x] is an action defined on the group H with x and s in H, and f is a function on H, that L_s[f] denotes the "example" action L_s[f](x) = f(x s) from line 112? Or could this L_s[f] be an arbitrary action defined on functions on H, with no relationship to the group action L_s[x]? Either way, please make this explicit. The next confusion arose in equation 14. Firstly, L_s is defined (line 145) as a mapping from X to X, where X might not be equal to the semi-group S. Then in equation 14, L_s[f] seems to be an action on functions on X. Is the action on functions L_s[f] defined by the group action L_s[x]? The "example" action L_s[f](x) = f(x s) cannot be used here because it depended on the fact that X = S. Perhaps this was just a typo, and L_s should have been defined on functions on X? Next confusion, equation 15. Throughout this equation, L_s and L_t represent a left-action on functions on X. Then, in the final expression, L_t is used to represent a left-action on functions on S. In particular, it seems to represent the "example" left-action L_s[f](x) = f(x s)? It might be helpful to provide a simple example of a left-action L_s[f] where f is a function on X and X ≠ S? Could you simply always assume that X = S for the definition of the semi-group convolution? This might have made things much clearer. You could also explicitly state that, when X = S, the action L_s[f] on functions on S is always the canonical left-action L_s[f](x) = f(x s)? For the example of rotations (line 131), the notation "R_s x" was confusing for me. I cannot immediately think of an N-D parameterization of an N-D rotation matrix such that associativity is satisfied: s t u = R_s (R_t u) = R_{R_{s} t} u This greatly confused me on the first pass because it seems like s and x are of different types in the expression "R_s x". Edit: I can see how this is true for 2D, where x = (cos theta, sin theta) and R can be constructed from cos and sin terms. Perhaps restrict the example to 2D? Or else, you could simply use L_s[x] = s x where s and x are rotation matrices? # Suggestions and minor corrections Line 62: There is a minus sign missing from the definition of G. Line 117: Should be "functions on X to functions *on* X"? Line 137: I didn't understand in what way the group action may "look different". Line 250: You might have meant to remove the word "whenever"?

Reviewer 2



I have concerns about the following aspects of the paper: (1) Related Work (missing citations and experimental baselines) The paper unfortunately fails to cite or compare to other recent neural network architectures that exploit scale space. For example: Multigrid Neural Architectures Tsung-Wei Ke, Michael Maire, Stella X. Yu. CVPR 2017 Feature Pyramid Networks for Object Detection Tsung-Yi Lin, Piotr Dollár, Ross Girshick, Kaiming He, Bharath Hariharan, Serge Belongie CVPR 2017 Multi-Scale Dense Networks for Resource Efficient Image Classification Gao Huang, Danlu Chen, Tianhong Li, Felix Wu, Laurens van der Maaten, Kilian Q. Weinberger ICLR 2018 While these particular example architectures do not enforce scale equivariance, that actually makes comparison to them especially important. We would like to know whether building in scale equivariance is actually a superior design. Perhaps merely using multi-scale representations suffices? (2) Novelty and presentation. The mathematical exposition seems unnecessarily long, especially as it boils down to using dilated convolution in order to achieve scale equivariance. This strategy for equivariance does not seem surprising and brings into question the overall degree of novelty. (3) Depth of Experiments By reporting only a few results on PCam and Cityscapes, against extremely simple baselines, it remains unclear whether the proposed design for scale-equivariance actually improves upon the state-of-the-art. At minimum, I would expect to see performance reported on widely used benchmark datasets for semantic segmentation, such as PASCAL or COCO, where object scale is important.

Reviewer 3



***Update**** My positive review was mostly from the point of view of work on group equivariance\covariance\invariance in neural networks, and the clarity of the presentation (hence the confidence rating as well). Nevertheless R2's thoughts have rightfully corrected my assessment. I am hopeful that the paper will also engage with work on multiscale CNNs in a better way. Needless to say that I really like the formalism; its transparency and potential to generalize to other domains and applications, but since the results are on images, adding a comparison to some of the baselines indicated should be considered more seriously. Not doing so might cloud some of the contributions of the paper and/or limit its dissemination in the vision community -- to which the paper also is clearly aimed at. While I was aware of some of the papers mentioned, thinking about them was easily missed while reading the paper through the GCNN lens, and I assume the same happened with the authors at least at some level. My opinion that this is a great paper does not change, but I will lower my rating one notch, as I do feel that the scholarship could be improved by heeding R2's concerns, which can help improve the paper even more. **************** Older version. Group equivariant neural networks have recently emerged as a principled approach to guide neural network design. The main idea in them is to explicitly bake in the symmetries of the input in the neural network architecture so that they are respected, thus obviating data augmentation for those set of transformations. Moreover, the use of group theoretic machinery also facilitates the use of techniques from harmonic analysis, making implementation clean and simple, especially for compact groups. One particular success story of GCNNs is that of simple discrete groups acting on images, like 90 degree rotations, flips and reflections (Cohen and Welling, 2016). Such networks are equivariant to not just translations as in classical CNNs, but also to these transformations in addition to translations. What is more, they can be implemented efficiently just in the "real space". Such simple generalizations of CNNs give a strong boost in image recognition, and as indicated above, without using explicit data augmentation of the nature described. It is however clear that another symmetry that matters in image recognition is that of scale. An image at different scales really represents the same object and thus the output should be invariant to scale. This is implemented in various ways in existing literature: Like considering various scales, training networks in parallel and voting; averaging images at different scales and feeding them in a network and various other permutations and combinations. Moreover, the role that scale plays in CNN architectures is not really understood. Use of atrous convolutions also shows this lack of understanding quite clearly by working so well. The main insight of this paper is that scale invariance could be incorporated in the GCNN framework seamlessly, while handling it in a similarly principled manner. This is because scale by itself forms a semi-group. While this causes some issues while defining correlation/convolution due to a lack of an inverse, but nonetheless a useful notion can be defined. The paper introduces the reader to the necessary literature on GCNNs, equivariance, scale spaces, and then describes their approach in a very clean manner. They do so by defining semi-group correlation, induced actions and the appropriate notion of convolution/correlation there and so on. Some subtle differences owing to the lack of inverse are also emphasized, such as the fact that the signal can be transformed but not the filter. The experimental results described are strong, and thorough, with even the quality of equivariance achieved clearly described, validating the approach. In short -- the paper is written in a very easy to read, and elegant manner. Proposes a generalization of group equivariant neural networks to semigroups, and uses this notion to incorporate scale invariance in image recognition. There aren't many comments that I can think of to improve the paper further. This short review is merely a reflection of the elegance and clear presentation of this paper.

[Author Response · NeurIPS 2019]

We would first like to thank the reviewers for especially detailed and high quality reviews.

**Main points**

**Notation, the overloaded use of $L_s$, R1**: Thanks for a very detailed break-down of issues with the notation. We
wanted to emphasize that the action $L_s$ is very general and so we used it in multiple definitions, as we felt that we did
not want to introduce too many symbols. That said, this may have had the opposite effect of being confusing. One way
we could resolve this would be to add the domain of the action as a superfix, for example $L_s^X : X \to X$ or $L_s^S : S \to S$
or $L_s^{F(X)} : F(X) \to F(X)$, where $F(X)$ is a space of functions on $X$. This will also disambiguate the difference
between an action in the input space and on the activations. We shall definitely spend more time explaining these
subtleties to readers. When it comes to specific examples of actions, as you suggest, it may just be better for us to use
specific notation, for instance $T_s$ for a translation or $R_s$ for a rotation.

**Examples of actions on whole signals, R1**: Indeed scale transformations (with bandlimiting) are actions on the whole
signal. Simple blurs are another example of a semigroup action on the whole signal. What we found interesting is
that while an action acts on the whole input, the first layer of convolution lifts that input on $X$ onto the semigroup $S$.
The induced action on these lifted activations always acts on just the domain, which as you point out allows us to use
pointwise nonlinearities. In our experiments with scale, we first lift to scale-space, which is why in the scale-equivariant
correlation that we ended up using, it appears we only ever act on the domain.

**Related work, R2**: Thank you for drawing these works to our attention. The paper "Multigrid Neural Architectures"
(Ke et al., 2017) is indeed close to ours, and we should be able to make a comparison in the camera-ready version of
our paper. "Feature Pyramid Networks for Object Detection" (Lin et al., 2017) and "Multiscale Dense Networks for
Resource Efficient Image Classification" (Huang et al., 2018) are also very interesting and have now been added to our
related work section.

**Novelty, R2**: By our own estimation and based on the reviews from **R1** and **R3**, we feel that the mathematical exposition
we provide contains an appropriate level of mathematical rigor. Furthermore, we feel it neatly dovetails the prior corpus
of work on group equivariance. Indeed the requirement for equivariance led to some not-so-obvious conclusions, for
instance, we only need to bandlimit once at the input of the network and not at every layer. The use of nonlinearities
actually increases the frequency content of each consequent activation, which from a signal-processing perspective
would indicate we need bandlimiting at each layer. Without the equivariance perspective, we would never have arrived
at our current solution. Indeed, in Dilated Residual Networks (Yu et al., 2017), there is no bandlimiting at all (we
presume it it probably learned in the first convolution). Moreover, maybe the use of dilated convolutions in the end "does
not seem surprising" as you state, but primarily we were interested in providing mathematically solid and principled
arguments behind why we make certain architectural decisions.

**Experiments, R2**: Of course we can perform more extensive empirical evaluations of the ideas set down in our paper,
and we intend for this to be the subject of follow up works. We believe this is essential to establish the utility of
equivariant methods in general. For the time being, we felt that one large dataset (Cityscapes, which was used in Dilated
Residual Networks), one medium sized dataset (PCam—digital histopathology can naturally benefit from scale), and an
introspective experiment were enough for a proof-of-concept. Furthermore, in related works there are no "standard
datasets". For instance, in Multigrid Neural Architectures test on CIFAR-100, a toy MNIST segmentation task, and
ImageNet, but in Feature Pyramid Networks, the authors look at COCO.

**More unstructured input spaces, R3**: Thanks for your very positive review, hence why this is the only point directed
at you. We do intend to extend the current work to other domains and to other kinds of semigroup action. Indeed
scale-spaces on graphs would be an interesting area to pursue, since diffusion equations are often deployed in Graph
CNNs.

**Minor points**

**Accuracy vs. training data amount, R1**: This is a good idea and we agree it sits inline with the motivations for
building in inductive biases, and thus we shall try to add this to the camera-ready version.

**Kanazawa et al, R1**: Thanks for pointing this out, we have now added it to the related work.

**Equation 22, R1**: Indeed $L_s'[x]$ is a right-action, we think the notation $R_s[x]$ would resolve this issue.

**Pretraining, R1**: This would be an interesting experiment to run, we shall try to fit it into the camera-ready paper.

**Equation 14, R1**: This is our mistake, $L_s$ should have been defined for functions on $X$, thanks for catching this.

**Rotation example, R1**: We shall make this example more explicit, to clear up confusion.

[Meta-Review · NeurIPS 2019]

This paper has resulted in a detailed discussion between the reviewers. The authors use the same techniques that have recently proved to be very popular for building neural networks that are equivariant to translations, rotations etc. to formulate a general theory of scale equivariant neural networks. I believe that the mathematical derivations are sound and that this is an interesting direction to pursue. However, at the end of the day the proposed algorithm is simple and similar to other "multi-scale" neural nets that have recently appeared in the literature. This makes one feel that the mathematical derivations are a little too meticulous, and instead the authors should have focused on conveying the intuition and providing more extensive and more convincing experimental results. The lack of the latter is particularly concerning given that comparable results published in the Vision literature are much stronger (by the authors admission they did not have the time/resources to perform similar experiments). On the whole, I felt that this paper is very borderline. In the camera ready version, I would strongly encourage the authors to update the paper by bringing it more inline with what has already been published in Vision, as explained by Reviewer 2. That would greatly improve its potential impact.